# Moderate Consumption of Beer and Its Effects on Cardiovascular and Metabolic Health: An Updated Review of Recent Scientific Evidence

**DOI:** 10.3390/nu13030879

**Published:** 2021-03-09

**Authors:** Ascensión Marcos, Lluís Serra-Majem, Francisco Pérez-Jiménez, Vicente Pascual, Francisco José Tinahones, Ramón Estruch

**Affiliations:** 1Institute of Food Science, Technology and Nutrition (ICTAN), Spanish National Research Council (CSIC), 28040 Madrid, Spain; 2CIBER Physiopathology of Obesity and Nutrition, Carlos III Institute of Health, 28222 Madrid, Spain; lluis.serra@ulpgc.es (L.S.-M.); md1pejif@uco.es (F.P.-J.); fjtinahones@hotmail.com (F.J.T.); restruch@clinic.cat (R.E.); 3Research Institute of Biomedical and Health Sciences, University of Las Palmas de Gran Canaria, 35016 Las Palmas, Spain; 4Lipid and Atherosclerosis Unit, Maimonides Biomedical Research Institute of Córdoba (IMIBIC), UGC of Internal Medicine, Reina Sofia University Hospital, University of Córdoba, 14004 Córdoba, Spain; 5Palleter Health Centre, CEU University—Cardenal Herrera, 12005 Castellón, Spain; pascual_vic@gva.es; 6Endocrinology, Nutrition, Diabetes and Obesity Unit, Institute of Biomedical Research in Málaga (IBIMA), Virgen de la Victoria Hospital, University of Málaga, 29010 Málaga, Spain; 7Department of Internal Medicine, Hospital Clinic, August Pi i Sunyer Biomedical Research Institute (IDIBAPS), University of Barcelona, 08036 Barcelona, Spain

**Keywords:** alcohol, moderate drinking, mortality, diabetes, obesity, osteoporosis

## Abstract

There is growing interest in the potential health-related effects of moderate alcohol consumption and, specifically, of beer. This review provides an assessment of beer-associated effects on cardiovascular and metabolic risk factors to identify a consumption level that can be considered “moderate”. We identified all prospective clinical studies and systematic reviews that evaluated the health effects of beer published between January 2007 and April 2020. Five of six selected studies found a protective effect of moderate alcohol drinking on cardiovascular disease (beer up to 385 g/week) vs. abstainers or occasional drinkers. Four out of five papers showed an association between moderate alcohol consumption (beer intake of 84 g alcohol/week) and decreased mortality risk. We concluded that moderate beer consumption of up to 16 g alcohol/day (1 drink/day) for women and 28 g/day (1–2 drinks/day) for men is associated with decreased incidence of cardiovascular disease and overall mortality, among other metabolic health benefits.

## 1. Introduction

In recent years, there has been an increasing interest in the potential health-related effects of moderate alcohol consumption. Although the harmful effects of excessive alcohol use are well established, the association of low-to-moderate alcohol consumption with health-related benefits is still controversial, since the results of available studies are not homogeneous and reaching clear conclusions is challenging. This lack of consensus is observed in alcohol consumption guidelines published in the last five years, which use different terminology (“risky drinking”, “moderate consumption”, or “low-risk drinking”) as well as different drinking thresholds [1,2,3,4,5,6] (Table 1). Furthermore, other variables, such as differences in concentrations of non-alcoholic components (i.e., polyphenols), may confound the beneficial effects of specific alcoholic drinks [7,8].

Beer is an alcoholic beverage frequently consumed in Europe. In 2018, the average yearly beer consumption in Europe was 72 L per capita, with a few countries (Czech Republic, Austria, and Germany) consuming more than 100 L per capita per year [9]. However, patterns of consumption differ across the region varying from predominantly meal-associated drinking in Mediterranean countries, to high rates of heavy episodic drinking in Central and Eastern Europe, and relatively frequent consumption both with and outside of meals in Central Western Europe [10].

Beer is mainly composed of water, but it is also rich in nutrients—carbohydrates, amino acids, minerals, vitamins, and polyphenols—resulting from a multi-step brewing and fermentation process [7,11,12,13]. Hop flowers, used as a bittering and flavoring agent [14], contain phenolic compounds, including prenylated flavonoids [15,16], which have been shown in vitro to have different antioxidant, anticarcinogenic, anti-inflammatory, oestrogenic, and antiviral biological activities [7,17]. Xanthohumol is the most abundant of these compounds and, in addition to potential bioactivity [7,18], it also inhibits platelet activation without increasing the bleeding risk [19]. Thus, brewing processes have been optimized to achieve the highest possible content of xanthohumol [20]. Regarding antioxidant content, ale beers have been reported to display a higher antioxidant activity than lager beers due to the higher fermentation temperature in the brewing process. However, despite these enrichment processes, controversy remains as to the bioavailability of the phenolic compounds in beer [21,22,23].

Alcohol content in regular beers varies between 3% and 6% alcohol by volume [11]. There is vast scientific literature on excessive alcohol consumption. Indeed, chronically high alcohol intake acts as a toxin to the heart and vascular system and may also exacerbate pre-existing heart disorders. However, low-to-moderate amounts of alcohol intake might have beneficial effects on the cardiovascular (CV) system, since it increases high-density lipoprotein cholesterol (HDL) and reduces arterial stiffness (both effects shown specifically with beer) [21,22,24,25], and also decreases fibrinogen, platelet activation and aggregation, as well as blood oxidative stress and inflammatory parameters [26,27,28]. The alcohol content of beer might also have an effect on glucose homeostasis [29]. Alcohol contributes to total calorie intake and may increase weight when consumed in excess [30,31]. Non-alcoholic components also contribute to the energy content of beer. Thus, Public Health England lists the mean energy content of alcohol-free beers at seven kilocalories/100 g [32]. Overall, 28% of the total monthly kilocalories contributed by beer among regular drinkers derive from its non-alcoholic ingredients [33].

Taken together, the biological activity of phenolic compounds in beer and the possible association of alcohol intake with mortality, CV risk [34,35,36], and glucose metabolism [37,38,39,40,41,42,43] may contribute to the putative health-related effects of moderate beer consumption. Conversely, excess beer consumption may be associated with weight increase and associated morbidities [30].

We conducted an analysis of all reviews, meta-analyses, and longitudinal, prospective, cohort studies published from January 2007 to April 2020 regarding beer consumption and its relationship with CVD and mortality risk, with the objective of evaluating the average intake of beer that could be considered “moderate” based on the reported consumption. Furthermore, we aimed to identify several differences shown in specific population subgroups in the selected studies.

## 2. Materials and Methods

In April 2020, a literature search of papers published after January 2007 was conducted using PubMed EMBASE, and through reference list cross-checking of previous meta-analyses, prospective clinical studies, and systematic reviews in humans, evaluating beer effects on health. The search strategy retrieved citations from databases containing the subject heading “beer” in combination with “health”, “cardiovascular”, “mortality”, “obesity”, “diabetes”, “young”, “women”, or “alcoholism”. The search terms were adapted for use with both bibliographic databases (Table 2).

Original eligible papers, based on their title/abstract, were obtained and reviewed to select those meeting the inclusion criteria: clinical studies in humans with prospective cohort design, plus systematic reviews and meta-analyses evaluating beer effects on health since 2007, and publications that authors considered could provide additional information on the subject. A total of 13 reviews (narrative or systematic reviews and/or meta-analysis), 9 prospective cohort studies, and 1 open-label, randomized, cross-over study were selected (Table 3).

## 3. Results

A summary of related studies is shown in Table 4. Two reviews referring specifically to beer, reported that moderate consumption (up to 55 g alcohol/day; i.e., 385 g/week) showed a beneficial effect on non-fatal CV events [34,57]. Both reviews found that the highest effect was associated with moderate beer or wine consumption, suggesting that the polyphenolic content of these beverages probably contributes to the observed CV benefits [34,57].

These data are in agreement with most findings from reviews/meta-analyses and cohort studies, which report a protective effect of moderate alcohol drinking for CVD compared to abstention, former drinking, or occasional drinking [22,34,47,49,56,57,60]. Only the review by Toma et al. does not support these conclusions; the authors suggest that this protective effect may be a confounder due to the inclusion of former drinkers in the non-drinkers group [48]. However, the prospective cohort study by Bell et al., conducted in almost 2 million people, took these potential confounders into consideration, and also found a positive effect of moderate drinking (112 g/week in women and 168 g/week in men) on CV risk [49]. Costanzo et al. found a negative (although not significant) association between spirits and vascular events, suggesting alcohol content, and not solely polyphenols, may also play a role in cardiovascular events [57]. Moreover, as Roerecke et al. [54] noted that distinctions between former drinkers and lifetime abstainers might not be sufficient for an accurate analysis, based on the heterogeneity of reasons underlying non-drinkers’ decision not to drink, which might further confound the results. Finally, drinking patterns may also play a key role in outcomes [54], since self-reported weekly intakes might include alcohol consumed during the weekend during a binge, which would be associated with worse CV results.

### Gender Differences

Due to the paucity of separate data for men and women, none of the beer-specific CVD studies stratified their conclusions by gender. The meta-analysis by Roerecke et al. [54] noted that women are more sensitive to the protective effect of moderate alcohol consumption, based on a previous meta-analysis [56], which showed a steeper J-curve for ischemic heart disease (IHD), mortality and morbidity in women than in men. The detrimental effect on CV risk of binge drinking seems to be lower in women than in men [49,60], and an increased risk of heart failure has been observed in abstemious women compared with moderate drinkers [49]. In addition, Snow et al. [60] observed that the beneficial effects of low and/or moderate usual consumption on CV risk were only evident in younger women (aged 18–34), whereas cardio-protection became evident at middle (aged 35–49) or older age (aged 50–64) in men. Overall, these gender differences may be due to hormonal changes over the life course or to a lower lifetime consumption of total alcohol.

## 4. Moderate Beer Consumption and Mortality

A summary of related studies is shown in Table 5. The systematic review by de Gaetano et al. [34] suggested that a J-shaped relationship also exists between beer consumption and all-cause mortality. The lowest mortality risk was observed in subjects with low to moderate alcohol consumption compared to abstainers or heavy drinkers, with the lowest risk at beer consumption of 84 g alcohol/week [34].

Studies on general alcohol consumption have similar conclusions. A prospective cohort study by Suadicani et al. found an association between wine consumption and all-cause mortality, with a consistent effect at 84 g alcohol/week, and a larger effect seen with higher consumption in men with non-O blood type [34,61]. However, a meta-analysis by Stockwell et al. [53] suggested that, when the necessary adjustments for study design characteristics are implemented, no association of moderate alcohol consumption with mortality is observed. Two later prospective studies, with well-adjusted variables, confirmed the association between lower risk of total and CV mortality [49,51] and moderate alcohol consumption defined by 168 g/week, 24 g/day for men and 112 g/week, 16 g/day for women in the Bell et al. study [49] or by 43–196 g/week for men and 43–98 g/week for women in the Xi et al. study [51].

### Gender Differences

In the EPIC study, beer consumption in women was more strongly related than wine consumption to overall mortality for amounts >21 g/week compared with the reference category (0.7–20.3 g/week) [69]. On the other hand, in the study by Xi et al., the protective effect of low and moderate alcohol consumption against all-cause and CV disease (CVD) mortality was more pronounced in women [51]. Thus, it seems that women may be both more sensitive to the protective effects against mortality of moderate beer intake and to the risk effects of higher amounts.

## 5. Moderate Beer Consumption and Obesity, Diabetes, and Osteoporosis

Table 6 details the studies on this subject. Although beer seems to have a direct effect on weight gain [52], and on waist circumference in men [59], there is not enough evidence to confirm whether moderate intake (<500 mL/day) is associated with general or abdominal obesity [55], although daily amounts ≥500 mL increase the risk of not losing weight [59]. In this regard, Padro et al., have reported that the moderate consumption of either alcoholic (30 g/day for men; 15 g/day for women) or non-alcoholic beer for four weeks did not increase the body weight of obese individuals [46]. Furthermore, moderate beer consumption was associated with increases in the anti-oxidative properties of high-density lipoprotein, which facilitate the efflux of cholesterol [46].

Based on the reviewed diabetes studies [43,50], moderate alcohol consumption may decrease diabetes risk in men. A meta-analysis of 13 prospective studies, with 397,296 participants, showed that wine consumption was associated with a significant reduction of the risk for type 2 diabetes mellitus (T2DM), with a pooled relative risk of 0.85, whereas beer or spirits consumption led to a slight trend towards a decreasing risk for T2DM (relative risk 0.96 and 0.95, respectively) [70]. Chronic alcohol consumption, however, is considered a risk factor for T2DM, which may be triggered by a deterioration in glucose tolerance, alterations in signalling of peptides involved in appetite regulation, and dysfunction and apoptosis of pancreatic β-cells [71,72].

Data on bone mineral density (BMD) and fracture risk have been less conclusive, probably due to the few studies available, and both relatively high and low levels of alcohol consumption have shown benefits for bone health. Thus, the consumption of both beer and wine at doses up to 60 g/day in men alone in the study by Yin et al. [58], and up to 13 drinks/week (182 g/week) in the study by Mukamal et al. [62], were shown to increase BMD and/or decrease risk of fracture in the elderly. Even very low levels of consumption were associated with a decreased fracture risk. Considering beer specifically, consumption of <1 beer/week (<14 g/week) in men and women was significantly associated with a lower risk of hip fracture (HR 0.66, 95%CI 0.44–0.99) [62]. Notably, low-alcohol beer consumption in women was associated with increased lumbar BMD [58], suggesting that, beyond the putative positive effect of alcohol on BMD, the non-alcoholic components of beer may also be involved. Other compounds present in beer (e.g., phytoestrogens such as 8-prenylnaringenin) act synergically with silicon to stimulate osteoblast cells, improve bone structure, and help remineralize bone and teeth [44]. The polyphenolic fraction, flavonoids, and the silicon content in beer may contribute to the positive effects on bone metabolism [45]. The protective effect of polyphenols has also been proven in human studies, where they reduced systolic and diastolic pressure and reduced lipoprotein cholesterol serum levels, among others [73]. The cardioprotective role of polyphenols in beer (traditional or alcohol-free) in particular has been reported in individuals with high cardiovascular risk [8,74].

### Gender Differences

With regard to obesity, the study by Schütze et al. [59] suggested that only men observe a risk for an increase in waist circumference (WC) with beer consumption of >500 mL/day. In women, beer-abstainers showed lower relative odds for WC gain compared with their very low-level drinking counterparts (1 to <125 mL/day), which was close to significance.

Similar gender differences were seen in the diabetes studies. Cullman et al. found that alcohol effect on glucose metabolism was different between men and women [43], depending on amounts of consumption and alcohol type; overall, in individuals with normal glucose tolerance, a decrease in T2DM risk was observed in occasional consumers of beer and wine vs abstainers among men, and in high consumers (≥192 g/week) of wine vs occasional consumers among women. This cohort study showed that men who were high consumers of beer and had baseline normal glucose tolerance had a significantly increased risk of developing abnormal glucose regulation (OR 1.63, CI 1.07–2.48 for pre-diabetes plus T2DM and OR 1.84, CI 1.13–3.01 for pre-diabetes) compared to occasional drinkers [43]. Men abstainers had a significantly higher risk of developing abnormal glucose regulation (OR 2.13, CI 1.03–4.39) than occasional beer drinkers, suggesting occasional beer consumption may be protective in men. Data for beer consumption in women were not provided in the Cullman et al. study. When considering individuals with normal glucose tolerance or pre-diabetes at baseline, the only significant difference found when using occasional drinking as a reference was the case of women with low consumption of total alcohol, who showed a decreased risk of T2DM (OR 0.41, CI 0.22–0.79). Most studies reviewed by Polsky et al. [50] also showed differences between men and women. In one study, a lower risk for T2DM was only observed in women who consumed alcohol (any quantity; no dose-relationship observed) compared to lifetime abstainers, but this was not found in men [75]. Another study showed that in men alone, a moderate alcohol consumption (10–14.9 g/day) was associated with a reduced risk of T2DM with respect to very low consumption (0.01–4.9 g/day), linked to wine consumption [76].

Regarding BMD, the study by Yin et al. [58] found that alcohol intake was positively associated only in men with an increase in the percentage of spinal and hip BMD after two years, whereas in women, lumbar spine BMD at baseline was positively associated with frequency of low-alcohol beer consumption (beta = 0.034 g/cm^2^ per category, *p* = 0.002).

## 6. Discussion

Despite the paucity of studies specifically exploring beer-related health effects, available data suggest that moderate beer consumption is associated with a decreased risk for non-fatal CV events and total mortality. For other health effects, such as those on general or abdominal obesity, study data have generally been inconclusive, although a recent small study suggests that moderate consumption of either alcoholic or non-alcoholic beer does not increase body weight in obese individuals [46]. Furthermore, moderate beer consumption has been associated with decreased diabetes risk (only in men), and with an increase in BMD, which lowers the risk of fracture in the elderly.

Although the available evidence supports the health benefits of moderate beer consumption in adults (aged ≥ 18 years), study heterogeneity makes it difficult to establish the precise quantity of beer needed to obtain these benefits. Different units of measurement (i.e., grams of alcohol or non-standardized drinks per day vs. standard drink units), and different definitions of consumption levels limit the direct comparison of these studies. Nevertheless, given the ranges of alcohol consumption associated with observed benefits in CVD (40–252 g/week for men; 21–210 g/week for women), mortality (75–196 g/week for men; 75–112 g/week for women), and diabetes, obesity, and osteoporosis (12–350 g/week for men; 12–210 g/week for women), a conservative upper limit of moderate beer drinking in men could be ≤196 g/week (approximately 1–2 beers per day). Available studies suggest that women may present a higher sensitivity to beer effects and, therefore, their upper limit of moderate consumption may be slightly lower at ≤112 g/week (approximately 1 beer per day). Notably, these values are similar to the low-risk drinking guidelines established by many countries (Table 1) [1,2,3,4,5,6]. Importantly, evaluating the overall effect of alcohol consumption on health is challenging, given it can be linked to improved benefits in CVD, as shown in this study, but it can also be associated with an increased risk of cancer [77]. Notably, moderate alcohol consumption varies between reports, and large-scale studies are warranted to adequately evaluate the role that specific ranges of alcohol consumption play in health.

Aside from gender differences, alcohol-associated health benefits might be modulated by intrinsic characteristics of populations, including their socioeconomic status [68] and/or diet and general lifestyle. In a well-established example, the Mediterranean diet—historically associated with high life expectancy and low CVD rates—is characterized by its high consumption of fresh foods, low consumption of animal fats, and low-to-moderate consumption of wine, generally with meals [78]. Indeed, the food pyramid recommended by the Spanish Society of Community Nutrition reflects the Mediterranean diet and includes fermented alcoholic beverages (wine, beer, and cider) among foods and drinks advised for an optional, occasional, and moderate consumption [3]. Importantly, this inclusion of alcoholic beverages in the food pyramid regards its consumption with meals, not alone. The inclusion of optional alcohol in the food pyramid is in agreement with data that shows that daily moderate beer consumption in the context of a Mediterranean diet is associated with favourable changes in the blood lipid profile [24].

The moderate consumption of alcohol ≤196 g/week (≤28 g/day or 1–2 beers daily) or ≤112 g/week (≤16 g/day or 1 beer daily) for men and women, respectively, has been associated with a variety of health benefits. However, it must be noted that weekly recommended amounts of alcohol should be spread across several days and not include episodes of heavy alcohol use or “binge drinking”, as irregular heavy drinking is associated with a higher risk of ischemic heart disease [79]. This episodic heavy drinking is defined by the World Health Organization as consumption of ≥60 g (approximately ≥6 drinks) per occasion [80] and by the United States National Institute on Alcohol Abuse and Alcoholism [81] as consumption of ≥5 drinks (male) or ≥4 drinks (female) in less than 2 h. While moderate regular drinking is associated with a lower risk of ischemic heart disease compared with abstention [54,82], excessive or binge drinking not only increases the risk of CV events, but also the risk of all-cause mortality [51,83]. Therefore, consumption of moderate amounts of alcohol should be always considered in the context of the Mediterranean lifestyle (moderate quantities of alcohol consumed as part of a meal), as a strategy to promote a more socially responsible consumption, avoiding excessive alcohol intake, often associated with Nordic European, as well as Central and Eastern Europeans and Anglo-Saxon alcohol consumers [10,84]. In a recent study using data from 123,219 men and women who were followed up to 34 years of age, the authors reported that the adherence of the participants to five low-risk lifestyle-related factors (never smoking, normal weight, regular physical activity, healthy diet, and moderate alcohol consumption) could prolong life expectancy at age 50 years by 14.0 and 12.2 years for female and male US adults, respectively, compared with individuals who adopted no low-risk lifestyle factors [85].

In developing this review, we considered the possibility that study sponsors might bias published results. Reported funding sources and conflict of interests (COIs) were assessed for every study selected for this review. Out of the selected studies, only three were directly funded by alcohol-related foundations [34,55,57], and COIs were reported by some of their authors. In addition, the sponsors of these three studies declared no intervention in the study execution and writing, and no differences in study results were observed regardless of funding source.

In conclusion, we consider that an approximate intake of 10–16 g alcohol/day (1 beer/day) for women and 20–28 g alcohol/day (1–2 beers/day) for men could be defined as moderate beer drinking, providing that the consumption is distributed throughout the week with no heavy episodic or “binge drinking” on a single occasion, especially during weekends. Moderate beer drinking decreases CV risk and overall mortality. In addition, moderate consumption decreases diabetes risk in men, increases BMD, lowering the risk of fracture in the elderly, and does not seem to be associated with general or abdominal obesity. Furthermore, moderate beer drinking should be considered within the context of mealtime consumption, as the custom in Mediterranean countries. Future studies should refine the quantity of beer considered as low-to-moderate consumption, which is the lowest risk level, and further determine the possible health benefits associated with moderate beer drinking.

Although research in this field is acquiring great interest and possible benefits are being found, the authors of this article insist on not recommending the consumption of alcohol in children, adolescents, pregnant women, adults under medication, or at work when using machinery (or driving). In addition, the consumption of alcohol must be always accompanied by meals and excess must be avoided.

## Figures and Tables

**Table 1 nutrients-13-00879-t001:** Low-risk drinking guidelines.

Country. Year Guidelines	1 SDU = g Pure Alcohol	Term	Daily ^a,b^ (g Alcohol)	Weekly ^a,b^ (g Alcohol)
**Spain. 2016** Socidrogalcohol consensus on alcohol in Primary Care [2]	1 SDU = 10 g Wine: 1 glass Beer: 1 beer (≈200 mL) Spirits: 25 g	Risky consumption (starting at)	Women: 20 g Men: 40–60 g	Women: 140 g Men: 280 g
**Spain. 2019** Update Dietary Guidelines for the Spanish population [3]	1 SDU = 10 g	Moderate consumption (upper limit)	Women: <20 g Men: <40 g	- -
**UK. 2016** UK Chief Medical Officers’ Low Risk Drinking Guidelines [4]	1 SDU = 8 g Wine: 1 glass (125 mL) (11% ABV)	Low-risk drinking (upper limit)	- -	Women: 112 g Men: 112 g
**USA. 2015** Dietary guidelines [5]	1 SDU = 14 g Wine: 5 fl oz or 147.9 mL (12% ABV) Beer: 12 fl oz or 354.9 mL (5% ABV) Spirits: 1.5 fl oz or 44.4 mL (40% ABV)	Moderate drinking (upper limit)	Women: 14 g Men: 28 g	- -
**Canada. 2018** Canada low-risk alcohol drinking guidelines [6]	1 SDU ^c^ ≈ 13 g Wine: 142 mL (12% ABV) Beer: 341 mL (5% ABV) Spirits: 43 mL (40% ABV)	Recommended limit	- -	Women: 130 g Men: 210 g
**37 countries. 2016** [1]	1 SDU = 8–20 g	Low-risk drinking (upper limits range)	Women: 10–42 g Men: 10–56 g	Women: 98–140 g Men: 150–280 g

ABV: Alcohol by volume; SDU: Standard drinking unit. ^a^ When amounts were expressed in number of SDUs, they were converted to grams. ^b^ Daily and weekly values are listed as published in respective guidelines. Weekly values may not reflect a week’s worth (7 days) of daily allowance. ^c^ 13 g is inferred from the different beverages considered as 1 SDU.

**Table 2 nutrients-13-00879-t002:** Literature search terms.

Data Base	Search Syntax	No. Articles
PubMed	((Beer[MeSH Major Topic]) and (“2007”[Date—Publication]: “2020/04/01”[Date—Publication])) and (health or mortality or cardiovascular or diabetes or obesity or women or men or gender or young or adolescent or age or alcoholism)	82
Filtered by:		
Clinical study		
Comparative study		
Multicenter study		
Observational study		
Randomized controlled trial		
Systematic reviews		
EMBASE	‘beer’/de and ‘beer’:ab,ti and ((‘health’/de or ‘mortality’/de or ‘cardiovascular’/de or ‘diabetes’/de or ‘obesity’/de or ‘female’/de or ‘male’/de or ‘sex’/de) and difference or ‘adolescent’/de or ‘young adult’/de or ‘alcoholism’/de) and ((article)/lim or (review)/lim) and ((adolescent)/lim or (young adult)/lim or (adult)/lim or (middle aged)/lim or (aged)/lim or (very elderly)/lim) and (humans)/lim and (clinical study)/lim and (2007–2017)/py	210

**Table 3 nutrients-13-00879-t003:** Papers selected for review.

Alcohol/Beer Paper	Study Type	Related Subject
Osorio-Paz et al. 2019 [44]	Review	CV
Sacanella et al. Nutr Hosp. 2019 [22]	Review	CV
Humia et al. Molecules 2019 [21]	Review	CV
Redondo et al. Nutr Hosp 2018 [45]	Review	CV and osteoporosis
Padro et al. Nutrients 2018 [46]	Prospective randomized cross-over	Obesity (metabolic syndrome)
Wood et al. Lancet. 2018 [47]	System. Review/Meta	CV and Mortality
Toma et al. Curr Atheroscler Rep. 2017 [48]	Review	CV
Bell et al. BMJ 2017 [49]	Prospective cohort	CV
Polsky et al. Curr Diab Rep. 2017 [50]	System. Review	Diabetes
Xi et al. J Am Coll Cardiol. 2017 [51]	Prospective population-based cohort	Mortality
de Gaetano et al. Nutr Metab Cardiovasc Dis. 2016 [34]	Review	CV and Mortality
Fresán et al. Nutrients. 2016 [52]	Prospective cohort	Obesity
Stockwell et al. J. Stud. Alcohol Drugs. 2016 [53]	System. Review/Meta	Mortality
Roerecke et al. BMC Med. 2014 [54]	System. Review/Meta	CV
Bendsen et al. Nutr Rev. 2013 [55]	System. Review/Meta	Obesity
Cullmann et al. Diabetic Medicine. 2012 [43]	Prospective cohort	Diabetes
Roerecke et al. Addiction. 2012 [56]	System. Review/Meta	CV
Costanzo S, et al. Eur J Epidemiol. 2011 [57]	System. Review/Meta	CV
Yin et al. Eur J Clin Nutr. 2011 [58]	Prospective cohort	Osteoporosis
Schütze et al. Eur J Clin Nutr 2009 [59]	Prospective cohort	Obesity
Snow et al. Age and Ageing. 2009 [60]	Prospective cohort	CV and Mortality
Suadicani et al. Alcohol. 2008 [61]	Prospective cohort	Mortality
Mukamal et al. Osteoporos Int. 2007 [62]	Prospective population-based cohort	Osteoporosis

CV. Cardiovascular.

**Table 4 nutrients-13-00879-t004:** Summary of the main cardiovascular (CV) studies.

Study Funding/COI ^a^	Design (Mean/Median Years of Follow-Up)	*n*	Categories of Consumption/Type of Drink	Variables	Reference Group (HR = 1)	Outcomes/Conclusions ^b^
Costanzo et al., 2011 [57] **Cervisia Consulenze** and Istituto Nazionale per la Comunicazione.	Systematic review-meta-analysis	12 prospective studies (*n* ranged from 1373 to 87,536) and 6 case-control studies (*n* ranged from 616 to 1746)	Wine, beer, and spirits	Fatal non-fatal CHD, CHD, CVD, AMI, stroke, CHD mortality, IHD mortality, CVD mortality AND/OR total mortality.		13 studies: J-shaped relationship for beer and CV risk. 16 studies: J-shaped relationship between wine intake and CV risk. 12 studies reporting separate data on wine or beer consumption: two closely overlapping dose-response curves.
Roerecke et al., 2012 [56] Global Burden of Disease Study and by the grant “Drinking Patterns & Ethnicity: Impact on Mortality Risks”, NIAAA	Systematic review-meta-analysis	44 observational studies including 957,684 participants	Lifetime abstainer. Occasional: 2.5–11.99 g/week; 12–23.99 g/week; 24–35.99 g/week / Alcohol in general	IHD	Lifetime abstainers	Cardioprotection was observed in all strata, and substantial heterogeneity was noted across studies. Wide confidence intervals observed particularly for average consumption of 1–2 drinks per day.
Roerecke et al., 2014 [54] The European Community’s Seventh Framework Programme—Addictions and Lifestyle in Contemporary Europe—Reframing Addictions Project.	Systematic review and meta-analysis	7 studies for the meta-analysis	For the meta-analysis: Current drinkers with an average alcohol consumption <30 g/day of pure alcohol with or without HED / Alcohol in general	IHD	Lifetime abstainers	Beneficial effect of low alcohol consumption without HED episodes as compared to life-time abstainers
Wood et al., 2018 [47] Various government, private, and pharmaceutical sources. / None declared.	Systematic review and meta-analysis	83 prospective studies including 599,912 participants	Current drinkers / Alcohol in general, also separate analyses for wine, beer, and spirits	Mortality, stroke, CHD, AMI, heart failure, fatal hypertensive disease, fatal aortic aneurysm	Lowest baseline alcohol consumption category (0–25 g/week)	Threshold for lowest risk of all-cause mortality was ~100 g/week. Association between alcohol consumption and total CVD risk showed higher HR for beer and spirits than for wine.
de Gaetano et al., 2016 [34] **Assobirra**, the Italian Association of the Beer and Malt Industries/Some authors were consultants for the Web Newsletter of Assobirra, or were on the board/received lecture fees from FIVIN, the Beer and Health Foundation, ERAB, or Cerveceros de España.	Systematic review	7 prospective studies (*n* ranged from 1373 to 87,526) and 4 case-control studies (*n* ranged from 937 to 1514)	Wine, beer, and spirits	Fatal non-fatal CHD, AMI, CHD, CHD mortality, AND/OR CVD mortality.		Some benefit of beer against CVD
Toma et al., 2017 [48]/ None declared	Review	1 case-control study (INTERSTROKE [63]; *n* = 26,919) and 1 prospective study (PURE; [64] *n* = 114,970)	INTERSTROKE: low-moderate alcohol use: ≤14 drinks/w in women and ≤21 drinks/w in men. / Alcohol in general	INTERSTROKE: any stroke, ischemic stroke, and hemorrhagic stroke PURE: mortality, CVD, myocardial infarction, stroke, and a composite of all outcomes, which also included alcohol-related cancers and injury.	INTERSTROKE: Non-drinkers or former drinkers. PURE: Non-drinkers	INTERSTROKE: Low-moderate alcohol use was associated with stroke (OR: 1.14; 95%CI 1.01–1.28), ischemic stroke (OR: 1.07; 0.93–1.23) and hemorrhagic stroke (OR: 1.43; 1.17–1.74). PURE: Current drinking was associated with reduced myocardial infarction (HR: 0.76; 95% CI 0.63–0.93). In addition, it was associated with a reduced composite outcome in high-income and upper-middle-income countries (HR 0.84 (0.77–0.92)), but not in lower-middle-income and low-income countries (HR 1.07 (0.95–1.21); *p*-interaction < 0.0001).
Bell et al., 2017 [49] National Institute for Health Research, Welcome Trust, the Medical Research Council prognosis research strategy Partnership and other government health-related agencies. / None declared	Prospective cohort (6 y)	1,937,360 (51% women)	Non-drinkers. Former drinkers. Occasional drinkers: drinks rarely or occasionally. Moderate: Men: 21 SDU/week or 3 SDU/day. Women: 14 SDU/w or 2 SDU/da. Heavy drinkers / Alcohol in general	12 common symptomatic manifestations of CVD. Aggregated CVD (all CV endpoints other than stable angina).	Moderate drinkers	**Non-drinking**: unstable angina (HR 1.33, 95% CI 1.21 to 1.45), myocardial infarction (1.32, 1.24 to1.41), unheralded coronary death (1.56, 1.38 to 1.76), heart failure (1.24, 1.11 to 1.38), ischemic stroke (1.12, 1.01 to 1.24), peripheral arterial disease (1.22, 1.13 to 1.32), and abdominal aortic aneurysm (1.32, 1.17 to 1.49). **Heavy drinking**: unheralded coronary death (HR 1.21, 95%CI 1.08 to 1.35), heart failure (1.22, 1.08 to 1.37), cardiac arrest/sudden coronary death (1.50, 1.26 to 1.77), transient ischemic attack (1.11, 1.02 to 1.21), ischemic stroke (1.33, 1.09 to 1.63), intra-cerebral hemorrhage (1.37, 1.16 to 1.62), peripheral arterial disease (1.35, 1.23 to 1.48); myocardial infarction (0.88, 0.79 to 1.00) and stable angina (0.93, 0.86 to 1.00).
Snow et al., 2009 [60] None declared	Prospective cohort (10 y)	1154 (574 women)	1 SDU: 13 g ethanol. Men: Light: 0.65–5.77 g/day; Moderate: 5.78–18.1 g/day; Heavy: >18.1 g/day. HED: ≥8 drinks/episode in past year Women: Light: 0.65–2.92 g/day; Moderate: 2.93–9.15 g/day; Heavy: >9.15 g/day. HED: frequency of ≥8 drinks/episode / Alcohol in general	CHD events; hypertension; Other CVD	Lifetime abstainers and occasional drinkers who consumed <0.05 drinks (<0.65 g) per day.	**Men for CHD events**: Heavy: HR: 0.28 (0.08–0.93) *p* = 0.037 in old men; HED: 4.13 (1.46−11.62) *p* = 0.0073 in middle-aged men. **Men for hypertension**: HED: HR 1.62 (0.99–2.63), *p* = 0.054 in old men. **Men for other CVD**: Light: HR 0.54 (0.28−1.04), *p* = 0.066 in old men; Heavy: HR 0.40 (0.19−0.85, *p* = 0.017 in old men. **Women for hypertension: Light**: HR 0.26 (0.07–1.01), *p* = 0.052 in young women. **Women for other CVD**: Light: HR 0.23 (0.08−0.65), *p* = 0.0057 and Moderate: HR 0.14 (0.05−0.45), *p* = 0.0009 in young women.

AMI: Acute myocardial infarction; COI: Conflict of interest; CV. Cardiovascular; CVD: Cardiovascular disease; CHD: Coronary heart disease; ERAB: European Foundation for Alcohol Research; FIVIN: the Foundation for Wine and Nutrition Research; HED: Heavy Episodic Drinking; HR: Hazard Ratio; IHD: Ischemic heart disease; OR: Odds Ratio; SDU: Standard drinking unit. ^a^ When funding is provided by industries and/or foundations that might represent a conflict of interest, it is written in bold. ^b^ Outcomes for prospective studies and meta-analyses, and Conclusions for reviews. Adjustments: Costanzo et al., 2011: All by age and 15 of the 18 studies, in addition, by one or more of the following: sex, race, education, marital status, country of birth, smoking, total alcohol intake, exercise intensity, depression score, frequent aspirin use, cholesterol, BMI, diabetes, hyper-lipidemia, cancer, physical activity, cohabitation, coffee, consumption of other beverage types, total daily energy and saturated fat intake, intake of vegetables, fruit, fish, saturated and trans fatty acids, socioeconomic status, history of heart dis-ease, AMI, hypertension, etc. Roerecke et al., 2012: Where needed, the effect sizes of reference categories were re-calculated to reflect abstainers as the reference category. Former drinkers were excluded from all analyses. Those consuming > 72 g/day were excluded from all analyses because of scarcity of data. Roerecke et al., 2014: All studies were adjusted for age and smoking status, five for education and other indicators for socio-economic status, and four each for BMI and marital status. Wood et al, 2018: Age, smoking status, history of diabetes. de Gaetano et al., 2016: All by age and 8 of the 11 studies, in addition, by one or more of the following: sex, race, education, marital status, country of birth, smoking, total alcohol intake, exercise intensity, depression score, frequent aspirin use, cholesterol, BMI, diabetes, hyperlipidaemia, cancer, physical activity, cohabitation, coffee, consumption of other beverage types, total daily energy and saturated fat intake, intake of vegetables, fruit, fish, saturated and trans fatty acids, socioeconomic status, history of heart disease, AMI, hypertension, etc. Bell et al., 2017: Adjusted for age (and age^2^), sex, socioeconomic deprivation, and smoking status. Snow et al., 2009: Adjusted for marital status, cigarette smoking status and educational level.

**Table 5 nutrients-13-00879-t005:** Summary of main mortality studies.

Study Funding/COI ^a^	Design (Mean/Median Years of Follow-Up)	*n* (Women)	Categories of Alcohol Consumption/Type of Drink	Variable/s	Reference Group (HR = 1)	Outcomes/Conclusions ^b^
de Gaetano et al., 2016 [34] **Assobirra**, the Italian Association of the Beer and Malt Industries/ Some authors were consultants for the Web Newsletter of Assobirra, or were on the board/received lecture fees from Fundación Cerveza y Salud, FIVIN, the Beer and Health Foundation, ERAB, or **Cerveceros** **de España.**	Systematic review		Wine, beer, and spirits	All-cause mortality		Evidence suggests a J-shaped relationship between alcohol consumption and total mortality, with lower risk for moderate alcohol consumers than for abstainers or heavy drinkers. Specific data on beer are not conclusive, although some results indicate a positive role of drinking beer in moderation (1 drink/day, about 12 g of ethanol) against mortality for any cause
	1 meta-analysis of 34 prospective studies [65]	Over 1 million adults	Low to moderate Women: 1 drink/day. Men: 2 drinks/day / Wine, beer, and spirits	All-cause mortality		Low to moderate consumption of alcohol significantly reduces total mortality, while higher doses increase it
	1 Prospective cohort [66] (12–18 y)	36,250 men	Wine and beer	CV death All-cause mortality	Non-drinkers	Moderate wine or beer drinking reduced the risk of CV death. Only moderate wine drinking was associated with lower all-cause mortality: RR: 0.67 (0.58 to 0.77; *p* < 0.001)
	1 Prospective cohort [67] (16.8 y)	7735 British men 40–59 y old	1 SDU: Half pint beer (8–10 g alcohol). Frequency: Non-drinkers; Occasional (1–2 SDU/month); Weekend drinkers; Daily or on most days. Quantity: 1–2, 3–6, >6 / Wine, beer, and spirits	All-cause mortality	Occasional drinkers	Regular beer drinking [HR: 0.84 (0.71 to 1.01)] showed no significant difference vs. occasional drinking
	1 Prospective cohort [68] Copenhagen City Heart Study (25 y)	14,223 adults	1 SDU: 1 bottle beer (12 g alcohol). Never, Hardly ever, Monthly, Weekly Daily: 1–2 SDUs Daily: >2 SDUs / Wine, beer, and spirits	All-cause mortality	Never beer drinkers	In men, monthly beer intake (RR: 0.86 (0.77 to 0.97)) was associated with lower mortality, and daily intake >2 beers (RR: 1.14 (1.02 to 1.27)) to increased risk. In women the associations were not statistically significant: Monthly beer intake (RR: 0.98 (0.88 to 1.08)), and daily intake >2 beers (RR: 1.31 (0.92 to 1.88)) At a medium education level, monthly beer intake was associated with lower risk (RR: 0.87 (0.77 to 0.97)), and at low [RR:1.20 (1.07 to 1.34) and medium education level (RR:1.18 (1.02 to 1.37)), >2 beers daily was associated with increased risk.
	1 Prospective cohort [69] (12.6 y)	380,395 adults (247,795 women)	For beer: Never. Light: 0.1–2.9 g/day, 3–9.9 g/day, 10–19.9 g/day, 20–39.9 g/day (only for men). ≥20 g/day (extreme for women) ≥40 g/day (extreme for men) / Wine and beer	All-cause mortality	Light consumers (0.1–2.9 g/day)	In women: Compared to low-level consumers, lifetime non-drinkers (HR: 1.06; 1.02 to 1.12), and consumers of beer at amounts ≥3 g/day displayed significantly higher overall mortality risk. In men: Lifetime non-drinkers (HR: 1.07; 0.98 to 1.16) and consumers of 3–9.9 g/day (HR: 1.04; 0.98 to 1.10) showed no significant differences compared to light consumers. Consumers of beer amounts ≥10 g/day displayed a significantly higher overall mortality risk.
Stockwell et al., 2016 [53] None declared	Systematic review/meta-analysis of 87 studies (13.4 y)	3998,626 adults	Abstainer. Former drinker. Occasional: <1.30 g/day. Low: 1.30 to <25 g /day. Medium: 25 to <45 g/day High: 45 to <65 g/day. Higher: ≥65 g/day / Alcohol in general	All-cause mortality	Abstainer OR occasional drinker	**Standard adjustment**: Significant protective effect for low-volume (RR: 0.86 (0.83 to 0.90); *p* < 0.0001) and occasional drinkers (RR: 0.84 (0.79 to 0.89); *p* < 0.0001) as compared with abstainers. Abstainers were at significantly higher risk (RR: 1.19 (1.12 to 1.27); *p* < 0.0001) as compared to occasional drinkers. **Full adjustment**: No significant protection was estimated for occasional (RR: 0.95 (0.85 to 1.05)), low-volume (RR: 0.97 (0.88 to 1.07)), or medium-volume drinkers (RR: 1.07 (0.97 to 1.18)) as compared with abstainers.
Xi et al., 2017 [51] None declared	Population survey data linked to mortality data (8.2 y)	333,247 adults	1 SDU: 14 g alcohol. Lifetime abstainers. Lifetime infrequent drinkers. Former drinkers. Current light drinkers. Moderate: >3 to ≤14 drinks/w for men or >3 to ≤7 drinks/w for women. Heavy drinkers. Binge drinking / Alcohol in general	All-cause, cancer, or CVD mortality.	Lifetime abstainers	All cause-mortality: Decreased for Light (HR 0.79 (0.76 to 0.82)) and Moderate (HR 0.78 (0.74 to 0.82)) drinkers. Increased in Heavy: HR: 1.11 (1.04 to 1.19) and binge (HR: 1.13 (1.04 to 1.23)) drinkers. CVD-specific mortality: Light: HR 0.74 (0.69 to 0.80); Moderate: HR 0.71 (0.64 to 0.78)
Bell et al., 2017 [49] National Institute for Health Research, Welcome Trust, the Medical Research Council prognosis research strategy Partnership and other government health-related agencies.	Prospective cohort (6 y)	1937,360 (51% women)	1 SDU ^c^: 8 g Non-drinkers. Former drinkers Occasional drinkers: drinks rarely or occasionally. Moderate: Men: 21 SDU/w or 3 SDU/day. Women: 14 SDU/w or 2 SDU/day Heavy drinkers / Alcohol in general	CV death and all-cause mortality	Moderate drinkers	Non-drinkers (former and occasional drinkers removed) had an increased risk of CV death (HR: 1.32 (1.27 to 1.38)) and all-cause mortality (HR: 1.24 (1.20 to 1.28)).
Suadicani, 2008 [61] The King Christian X’s Foundation, The Danish Medical Research Council, The Danish Heart Foundation, and The Else & Mogens Wedell Wedellsborg Foundation.	Prospective cohort (16 y)	3022 Caucasian males 53–74 y old	1 SDU: 10–12 g ethanol / Wine, beer, and spirits	All-cause and IHD-related death within the different blood phenotypes	Alcohol abstainers (comparison only for wine drinkers)	For beer, the median (P_20_, P_80_) number of drinks/week among those with the non-O phenotype was significantly higher in those who died (overall mortality): 10.5 (0, 15.5) vs 7.5 (0, 10.5); *p* ≤ 0.001. The effect of wine intake on all-cause mortality among middle-aged and elderly men may depend on ABO phenotypes. Among non-O phenotype, drinking 1–8 drinks/w: HR: 0.8 (0.7 to 1.8) and drinking >8 drinks/w: HR: 0.7 (0.6 to 0.98)

CV. Cardiovascular; CVD: Cardiovascular disease; ERAB: European Foundation for Alcohol Research; FIVIN: the Foundation for Wine and Nutrition Research; HR: Hazard Ratio; IHD: Ischemic heart disease; OR: Odds Ratio; SDU: Standard drinking unit; (to): 95%CI ^a^ When funding is provided by industries and/or foundations that might represent a conflict of interest, it is written in bold. ^b^ Outcomes for prospective studies and meta-analyses, and Conclusions for reviews. ^c^ Since Bell et al. follow UK guidelines, 1 SDU was assumed to be 8 g alcohol. Adjustments: de Gaetano et al., 2016: (A) 1 Prospective cohort [67]; For age, social class, smoking, physical activity, body mass index, lung function, evidence of CHD on questionnaire, diabetes, and regular medication. (B) 1 Prospective cohort [68]; For other types of alcohol, sex, smoking, body mass index, physical activity in leisure time, cohabitation, and education. (C) 1 Prospective cohort [69]; For age at recruitment, BMI and height, former drinking, time since alcohol quit-ting, smoking status, duration of smoking, age at start smoking, educational attainment, and energy intake. In women also for menopausal status, ever use of replacement hormones and number of full-term pregnancies. Stockwell et al., 2016: Standard adjustment for between-study variation in covariates: Former drinker, Occasional (<1.30 g/day), Low volume (1.30 to <25 g /day), Medium volume (25 to <45 g/day), High volume (45 to <65 g/day), Higher volume (65 g/day), All drinkers combined. Full adjustment for study characteristics: median age at intake, sex, Caucasian/non-Caucasian, drinking measure adequacy, former drinker bias, and occasional drinker bias. Xi et al., 2017: Model 1: Adjusted for age, sex, and race or ethnicity. Model 2: Additional adjustments for education, marital status, body mass index, physical activity, smoking, and diabetes) Bell et al., 2017: HRs adjusted for age (and age 2), sex, socioeconomic deprivation, and smoking status. Suadicani, 2008: Age adjusted (only for wine drinking).

**Table 6 nutrients-13-00879-t006:** Summary of main obesity, diabetes, and osteoporosis studies.

Study Funding/COI ^a^	Design (Mean/Median Years of Follow-Up)	*n* (Women)	Categories of Alcohol Consumption/Type of Drink	Variable/s	Reference Group (HR = 1)	Outcomes/Conclusions ^b^
Fresan et al., 2016 [52] The Spanish Ministry of Health, the Navarra Regional Government, and the University of Navarra.	Prospective cohort (4 y)	15,765 adults	Beverages groups: Water, low/non-caloric beverages (diet soda beverages, coffee without sugar), milk, juice, and sugared coffee (dairy products, juices, coffee with sugar). Occasional consumption (SSSBs and spirits). Wine, beer	Change in BW and new-onset obesity	No substitution	Substitution of one beer with one serving of water per day at baseline was related to a lower incidence of obesity (OR 0.81, 95%CI 0.69 to 0.94 and OR 0.84, 95%CI 0.71 to 0.98, when further adjusted for the consumption of other beverage groups) and to higher weight loss (−328 g, 95%CI −566 to −89).
Bendsen et al., 2013 [55] **The Dutch Beer Institute** (funded by the Dutch Brewers)/ Three of the authors are employed by or are board members of the Dutch Beer Institute.	Systematic review of 35 observational studies and 12 experimental studies Meta-analyses: 14 observational studies (11 cross-sectional and 3 prospective) included in dose-response graphs. 10 intervention studies (6 beer vs non-alcoholic beer and 4 beer vs control) included in quantitative synthesis		1 SDU beer = 330 mL, 4.6% alcohol = 12 g/drink. / Beer	BW increase, BMI, and abdominal obesity (WC and WHR)	Control: Non-drinkers or in the absence of non-drinkers, the group with the lowest beer intake Low or non-alcoholic beer	Dose-response graphs: High beer intake (>4 L/w) was associated with a higher degree of abdominal obesity in men. Quantitative synthesis: High beer consumption (about 1000 mL/day; 5% alcohol) did not result in increased BW compared with control groups but did result in increased BW compared with low- or non-alcoholic beer groups (mean difference 0.73 kg, 95% CI: 0.53 to 0.92; z = 7.39, *p* < 0.0001, I2 = 0%)
Schütze et al., 2009 [59] The German Cancer Aid, the German Federal Ministry of Education and Research and the European Union.	Prospective cohort (8.5 y)	20,625 (12,749 women)	WOMEN: No beer. Very light: >0 to <125 mL/day. Light: ≥125 to <250 mL/day. Moderate: ≥250 mL/d MEN: No beer. Very light: >0 to <250 mL/day. Light: ≥250 to <500 mL/day. Moderate: ≥500 to <1000 mL/day Heavy: ≥1000 mL/day / Beer	WC change BW change	Very light	MEN: Moderate beer consumption showed significant lower relative odds for WC loss (OR 0.44, 95%CI 0.24 to 0.80) WOMEN: Although beer-abstaining women showed significantly lower relative odds (OR.0.88; CI 0.81, 0.96) for WC gain compared with their very-low-level-drinking counterparts, significance was lost once the model was adjusted by HC change; however, the new OR was on the border of significance (OR.0.91; CI 0.83, 1.00)
Padro et al., 2018 [46] **Fundacion Cerveza y Salud, Madrid, Spain; The European Foundation for Alcohol Research**; Spanish Ministry of Economy and Competitiveness of Science; Institute of Health Carlos III.	Open-label, prospective randomized, two-arm, longitudinal cross-over	36 (15 women)	WOMEN: 330 mL/day normal or non-alcoholic beer (15 g/day or 0 g/day alcohol) MEN: 660 mL/day normal or non-alcoholic beer (30 g/day or 0 g/day alcohol) / Beer	BMI T2D Lipid Profile		Moderate beer consumption (traditional or alcohol-free) does not increase body weight in obese healthy individuals or have negative effects on the vascular system. Moderate consumption was associated with reduced risk of dyslipidemia, increased anti-oxidative properties of high-density lipoprotein, and increased efflux of cholesterol.
Polsky et al., 2017 [50] None declared	Systematic Review of 96 studies	18 studies included more than 10,000 subjects each.	Alcohol in general			Moderate alcohol consumption generally reduces diabetes risk.
Cullman et al. 2012 [43] The Swedish Research Council; the Swedish Diabetes Association; the Swedish Council of Working Life and Social research; and Novo Nordisk Scandinavia.	Prospective cohort (8–10 y)	5128 adults (3058 women) with normal glucose tolerance and 111 (41 women) with pre-diabetes. 35–56 y old	Abstainers Total alcohol Occasional: 0.01–1.49 g/day in women, 0.01–6.79 g/day in men. Low: 1.50–4.71 g/day in women, 6.80–13.01 g/day in men. Medium: 4.72–8.75 g/day in women, 13.02–22.13 g/day in men. High: ≥8.76 g/day in women, ≥22.14 g/day in men Wine Occasional: ≤0.32 g/day in women, ≤0.99 g/day in men. Medium: 0.33–1.65 g/day in women, 1–4.99 g/day in men. High: ≥1.66 g/day in women, ≥5 g/day in men Beer (only in men) Occasional: ≤0.99 g/day. Medium: 1–4.99 g/day. High: ≥5 g/day / Wine, beer and spirits	PreD T2D PreD + T2D	Occasional drinkers	Normal glucose tolerance at baseline MEN: High alcohol: Higher risk of preD + T2D (OR 1.42, 95% CI 1.00–2.03). High beer: Higher risk of preD + T2D (OR 1.63, 95% CI 1.07–2.48) and higher risk of preD (OR 1.84, 95% CI 1.13–3.01) Abstainers vs occasional wine or beer drinkers: Higher risk of preD + T2D (OR 2.01, 95%CI 1.01–3.98 and OR 2.13, 95%CI 1.03–4.39, respectively). WOMEN: High wine: lower risk of preD (OR 0.66, 95% CI 0.43–0.99) Normal glucose tolerance or preD at baseline WOMEN: Low alcohol: Lower risk of T2D (OR 0.41, 95% 0.22–0.79). Medium wine: Lower risk of T2D (OR 0.46, 95%CI 0.24–0.88)
Yin et al., 2011 [58] National Health and Medical Research Council of Australia, Tasmanian Government and Royal Hobart Hospital Acute Care Programme.	Prospective cohort (2 y)	862 (49% women) Mean age 63 y, range 51–81	1SDU: 10 g alcohol Frequency: Never, <once a month, 1–3 days/month, 1/2/3/4/5/6 days /wk, every day. Amount 30mL spirits: 1 glass. 1 can beer: 2 glasses. 1 bottle wine (750 mL): 6 glasses. 1 bottle sherry (750 mL): 12 glasses. g/day / Wine, beer, and spirits	BMD change		Total alcohol intake in men positively predicted change in BMD at the lumbar spine and hip (beta = 0.008% and 0.006% per year per gram of alcohol intake, *p* < 0.05). The frequency of drinking red wine was positively associated with percentage change in BMD at the lumbar spine in men (beta= 0.08% per year per class, *p*= 0.048). At baseline, lumbar spine BMD was positively associated with frequency of low-alcohol beer drinking in women (beta = 0.034 g/cm(2) per category, *p* = 0.002).
Mukamal et al., 2007 [62] The National Heart, Lung, and Blood Institute. The National Institute on Ageing.	Prospective population-based cohort study (12 y If no hip fracture 7.5 y If hip fracture)	5865 ≥60 y	1 SDU: 12-ounce can or bottle of beer, 6-ounce glass of wine, and 1 shot of liquor. 1 SDU^c^ = 14 g Categories Long-term abstainers, former drinkers, <1 drink/w, 1–6 drinks/w, 7–13 drinks/w, ≥14 drinks/w / Wine, beer, and spirits	Hip fracture BMD	Long-term abstainers	Strong, graded, positive relationship between greater alcohol consumption and greater BMD up to 13 drinks/week. U-shaped relationship between alcohol intake and risk for hip fracture (quadratic trend: *p* = 0.02), with lower HRs in intermediate drinking categories. Drinking <1 beer/w showed a significantly lower risk of hip fracture (HR 0.66, 95%CI 0.44–0.99).

BMD: Bone Mineral Density; BMI: Body mass index; BW: Body weight; COI: Conflict of interest; HR: Hazard Ratio; OR: Odds Ratio; PreD: Pre-diabetes; SDU: Standard drinking unit; SSSBs: Sugar-sweetened soda beverages; T2D: Type 2 diabetes; WC: Waist circumference; WHR: Waist-to-hip ratio. ^a^ When funding is provided by industries and/or foundations that might represent a conflict of interest, it is written in bold. ^b^ Outcomes for prospective studies and meta-analyses, and Conclusions for reviews. ^c^ 14 g are inferred from the amounts of the different beverages constituting 1 SDU and the American guidelines. White rows: Diabetes studies; Light grey rows: Obesity studies; Dark grey rows: Osteoporosis studies. Adjustments: Fresan et al., 2016: Sex, age, age squared, baseline BMI, physical activity, smoking habit, personal and family history of obesity, following a special diet, adherence to the Mediterranean dietary pattern, snacking between meals, weight change during the five years prior to baseline, and total energy intake from other sources than the exchanged beverages. When the analyses were carried out for group of beverages, an additional adjustment for servings per day of other groups was conducted. Schütze et al., 2009: Age, physical activity, smoking, change in smoking status, alcohol in g/d from other alcoholic beverages, education, waist circumference at baseline, total non-beer energy intake, incident diseases during follow-up time and for women, additionally for menopausal status. Further adjustment for concurrent changes in body weight and hip circumference. Cullman et al. 2012: Age, BMI, tobacco use, physical activity, family history of diabetes and education (and the other beverage types when analyzing a specific beverage) Yin et al., 2011: Age, body mass index, physical activity, medication, calcium intake, and smoking. Mukamal et al., 2007: Age, sex, race, current weight, and height. Further adjustment: Smoking status, difficulty arising from a chair or bed, arthritis, diabetes, hypertension, clinical cardiovascular disease, previous cancer, weight in early teens, leisure-time physical activity, visual problems, MMSE score, and use of estrogens, thiazide-type diuretics, and thyroid agents.

## Data Availability

No new data were created or analyzed in this study. Data sharing is not applicable to this article.

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
