# Peer review of "Moderate Consumption of Beer and Its Effects on Cardiovascular and Metabolic Health: An Updated Review of Recent Scientific Evidence"

_nutrients, 2021, doi:10.3390/nu13030879_

Round 1

Reviewer 1 Report

Major points:

  • lines 85-56, Author should explain better the rationale of defining a “moderate” average intake of beer considering mainly CVD and mortality risks and omitting other kind of risks, like cancer (https://pubmed.ncbi.nlm.nih.gov/25422909/). Please note that “the relative risks of cancer are similar for beer, wine and spirits” (ref. 34) and that “recommendations for clinical practitioners remain challenging because of the apparent simultaneous beneficial and detrimental effects from on average low alcohol consumption” (ref. 50).
  • Abstract (lines 27-29): is the conclusion valid also for alcohol-free and low-alcohol beers? Please discuss thoroughly in the text (not only with respect to lumbar spine BMD or body weight).
  • Line 116 and the entire paragraph and elsewhere: the source of the value “<0.65” (ref. 55) must be specified more precisely (or omitted) and pertinence of cited references with the assumption stated must be checked. For example, ref. 23 refers, for low-risks amounts, to curve-extrapolated indications from ref. 52 and misses such value. Moreover, “effect of moderate alcohol drinking” is inappropriate and misleading and should be completed with the type of alcoholic drink tested in each of the cited references. In fact, as long as the meta-analysis reported in ref. 52 demonstrates a NEGATIVE, though non-significant, association between spirit intake and vascular events, the beneficial effects of alcohol itself are very questionable. Moreover, authors of ref. 45 themselves conclude that “a more nuanced approach to the role of alcohol in prevention of cardiovascular disease is necessary”. Regarding ref. 55, reported analyses were not conducted separately for alcohol type, thus assumptions for beer must be considered carefully. Ref. 60 concludes “cardioprotective association between alcohol use and ischaemic heart disease cannot be assumed for all drinkers, even at low levels of intake”, which is quite different “than protective effects of moderate alcohol drinking vs. abstainers ecc.”. In fact, as stated also in WHO’s “Global status report on alcohol and health 2018” (https://www.who.int/publications/i/item/9789241565639), “the relationship between alcohol and the onset of ischaemic heart disease or ischaemic strokes is complex; people who consume low-to-moderate amounts of alcohol and do not engage in irregular heavy drinking have a lower disease risk, while people who engage in irregular heavy drinking or who consume higher volumes of alcohol have a higher disease risk”. Author should consider this position of WHO in the discussion too (lines 275-276). Finally, ref. 61 concludes “data support limits for alcohol consumption that are lower than those recommended in most current guidelines”.

Minor points

Lines 51-63: the paragraph is useless unless, for example, the roles of phenols in health-related effects are considered in the definition of the “moderate” amounts of alcohol in beer.

Lines 78-81: check sentence. A subject appears to be missing.

Table 2: check “3000”.

Line 109: check "although".

Table 4: the table must include all types of drink. To easier comprehension, maybe COIs could be placed at the bottom as table notes (please check also the suitability of "none declared" statements) and reviews could be mentioned only in the text. For Costanzo et al, only divergent results on wine could be cited. Verify if “adjustments” can be merged or moved in the footnotes. IHD meaning is missing in the table notes. Also for De Gaetano et al. check if “adjustments” can be merged or moved.

Table 5 and 6: same as table 4 for readability comments, COI and adjustments could possibly be moved in the footnotes and only significant HR values could be kept in the tables.

Lines 179-184: authors should also consider effects of chronic consumption and induction of insulin resistance (https://pubmed.ncbi.nlm.nih.gov/22540046/)

Lines 211-220: what about the comparisons with abstainers?

Lines 235-237: the “recent small study” must be cited

Author Response

Please, see attachment

Reviewer 2 Report

This is an interesting and well written review on an important topic. The manuscript is comprehensive and includes all the major studies in the field.

Reviewer 3 Report

Minor comments

1) Please check editing of the references.

2) Line 196-200 authors focused on the effect of the polyphenols on bone metabolism and teeth health, which in my opinion its not the main outcome/focus of the study. Authors should instead report the polyphenolic content of beer (not only flavonoids but also phenolic acids) and briefly discuss the health effects of these polyphenol classes toward human health (there are meta-analysis investigating the effect toward CVD, CVD-related mortality, and hypertension).
